# Light Field View Synthesis Using the Focal Stack and All-in-Focus Image

**DOI:** 10.3390/s23042119

**Published:** 2023-02-13

**Authors:** Rishabh Sharma, Stuart Perry, Eva Cheng

**Affiliations:** School of Professional Practice and Leadership, University of Technology Sydney, Ultimo, NSW 2007, Australia

**Keywords:** light field synthesis, focal stack, depth map

## Abstract

Light field reconstruction and synthesis algorithms are essential for improving the lower spatial resolution for hand-held plenoptic cameras. Previous light field synthesis algorithms produce blurred regions around depth discontinuities, especially for stereo-based algorithms, where no information is available to fill the occluded areas in the light field image. In this paper, we propose a light field synthesis algorithm that uses the focal stack images and the all-in-focus image to synthesize a 9 × 9 sub-aperture view light field image. Our approach uses depth from defocus to estimate a depth map. Then, we use the depth map and the all-in-focus image to synthesize the sub-aperture views, and their corresponding depth maps by mimicking the apparent shifting of the central image according to the depth values. We handle the occluded regions in the synthesized sub-aperture views by filling them with the information recovered from the focal stack images. We also show that, if the depth levels in the image are known, we can synthesize a high-accuracy light field image with just five focal stack images. The accuracy of our approach is compared with three state-of-the-art algorithms: one non-learning and two CNN-based approaches, and the results show that our algorithm outperforms all three in terms of PSNR and SSIM metrics.

## 1. Introduction

In conventional photography, only limited information from the light passing through the camera lens is captured. In general, each point in the captured images is the sum of the light ray intensities striking that point, not the total amount of light traveling along different directions that contribute to the image [1]. In contrast, light field imaging technology can capture rich visual information by representing the distribution of light in free space [2], which means capturing the pixel intensity and the direction of the incident light. Light fields can be captured using either an array of cameras [3] or a plenoptic camera [4]. However, capturing a light field with high spatial and angular resolution is challenging because plenoptic cameras have a spatial-angular resolution trade-off [4], and the set-up for a dense camera array is complex.

The additional dimensions of data captured in light field images enable generating images with extended depth of field and images at different focal lengths using ray-tracing techniques. Light field images also allow researchers to explore depth estimation techniques such as depth from defocus and correspondence, stereo-based matching, and using epipolar images from a single light field image. Depth estimation is crucial in computer vision applications such as robot vision, self-driving cars, surveillance, and human–computer interactions [5]. Stereo-based matching algorithms for light field images are mainly based on energy minimization and graph cut techniques. Kolmogorov and Zabih [6] combine visibility and smoothness terms for energy minimization. On the other hand, Bleyer et al. [7] consider the pixel appearance, global MDL (Minimum Description Length) constraint, smoothing, soft segmentation, surface curvature, and occlusion. However, the stereo-based depth estimation methods suffer from ambiguities while dealing with noisy and aliased regions. The narrow baseline makes it difficult for these algorithms to solve the occlusion problem [8]. In their work, Schechner and Kiryati [9] study the advantages and disadvantages of depth from defocus and correspondence techniques. While depth from stereo and depth from defocus and correspondence techniques can also be used for non-light field images, depth from epipolar images can only be used for light field images. Epipolar images (EPI) are formed by stacking the light field sub-aperture images in the horizontal and vertical directions, and a slice through this 4D representation reveals the depth of the pixels in terms of the slope of the line. However, due to the noise present in images, basic line fitting techniques do not produce robust results [10]. Zhang et al. [8] use a spinning parallelogram operator and estimate the local direction of a line in an EPI. They consider pixels on either side of the line separately to avoid the problem of the inconsistency of the pixel distribution. This also makes their depth estimation algorithm more robust to occlusions and noise compared to the EPI algorithm presented by Criminisi et al. [11] and Wanner and Goldluecke [12]. The techniques explored for depth estimation of light field images form the building blocks for light field image reconstruction and synthesis.

Light field reconstruction and synthesis algorithms can solve the problem of lower spatial resolutions for hand-held plenoptic cameras, and the ability to convert 2D RGB images to 4D light field images will change how we perceive traditional photography. Many algorithms that propose light field reconstruction techniques use a sparse set of light field views to reconstruct novel views [13,14,15,16]. However, the input data for these algorithms are a set of sub-aperture images, which is not easy to capture because the camera needs to be moved to capture the sub-aperture views using a 2D camera, and this is time-consuming and introduces issues of alignment. In contrast, for capturing a focal stack and all-in-focus images, we only need to change the focus and aperture of the camera without physically moving the camera. Thus, these algorithms cannot be used for light field synthesis but can be used for either increasing the spatial and angular resolution of light field images or for light field image compression. In this paper, the term ‘light field synthesis’ is considered an approach to creating an entire light field image with fewer input images, and we are not trying to synthesize the views with a large or small baseline. We are only focusing on mimicking the light field sub-aperture views using characteristics of the EPI of the light field images while using the central all-in-focus image and depth map.

On the other hand, light field synthesis can also be classified into two main categories: non-learning-based and learning-based approaches. Non-learning-based light field synthesis algorithms use a deterministic approach, where the same rules are used to synthesize the view for every image. These synthesis algorithms can be further divided into two categories based on the input data used: focal stack images or stereo image pairs. Stereo image pair-based algorithms either use micro-baseline image pairs or an image pair with a large baseline. Zhang et al. [17] propose a micro-baseline image pair-based view synthesis algorithm. Since the disparity between the stereo pair is small, the images can be captured by vibrating a static camera. Chao et al. [18], on the other hand, use a large baseline stereo pair. As the algorithm uses a large baseline, the horizontal views are synthesized by interpolating the stereo pair. In contrast, the main advantage of using focal stack images for light field synthesis is that the focal stack images can provide information to fill the gaps created near occluded regions in the synthesized views; key recent algorithms are in [14,19,20].

Learning-based light field synthesis uses a probabilistic approach, where the training input images are used to fit a model that can map the output. The two main drawbacks of learning-based light field synthesis approaches are that, first, a large amount of training data are required to train the network adequately; second, that the algorithm’s accuracy directly relates to the training data quality. Some learning-based algorithms [18,21,22] synthesize the entire 9 × 9 sub-aperture light field images using two, four, and one input image, respectively. Although these algorithms use fewer images as input, the Convolutional Neural Network (CNN) must be trained on a significant amount of training data to ensure high accuracy.

Our work thus uses a non-learning-based light field synthesis approach that does not require training data, yet can use varying sizes of focal stacks to synthesize light field images with high accuracy. We use the focal stack images and the all-in-focus image to synthesize the light field image: first, estimating a depth map using depth from defocus; then, refining the depth map using maximum likelihood for pixels estimated to incorrect depths. The depth map and the all-in-focus image are then used to synthesize the sub-aperture views and their corresponding depth maps by shifting the regions in the image. In our work, we show that, without using input images with a large baseline, we can still mimic the apparent disparity of the objects at different depths in sub-aperture views by only using the depth map and all-in-focus image. The missing information in the synthesized views is only where the foreground object moves more than the background objects for occluded regions. We handle these occluded regions in the synthesized sub-aperture views by filling them with the information recovered from the focal stack images. We compare our algorithm accuracy with one non-learning and two learning-based (CNN) approaches, and show that our algorithm outperforms all three in terms of PSNR and SSIM metrics.

### Our Contributions

Our proposed non-learning-based light field synthesis approach improves synthesis accuracy by:Synthesizing high-accuracy light field images with varying sizes of focal stacks as input, filling the occluded regions with the information recovered from the focal stack images;Using the frequency domain to mimic the apparent movement of the regions at different depths in the sub-aperture view, ensuring sub-pixel accuracy for small depth values.

## 2. Related Work

A well-known method for image synthesis using intermediate views of a scene is image interpolation [23]. View interpolation is then the process of estimating intermediate views given a set of images of the scene from different viewpoints. In an earlier key work on image synthesis [24], pixel correspondences are established using the range data and the camera transformation parameters between a pair of two consecutive images. A pre-computed morph map is then used to interpolate the intermediate views. In their work, they also talk about the holes that are generated in the estimated intermediate views. As the foreground regions in the estimated views move more than the background regions, these holes are filled by interpolating the adjacent pixels near the holes. However, this causes the filled regions to be blurry.

Building on these early image synthesis views, non-learning light field synthesis approaches either use a sparse set of perspective views to synthesize the view inside of the image baseline, or use a focal stack or the central view to extrapolate the perspective views. As mentioned previously, light field synthesis can be more broadly classified into two main categories: non-learning based and learning based approaches. Whilst we use focal stack images in a non-learning approach, below, we review both learning and non-learning light field synthesis methods as our results are compared to both types of approaches.

### 2.1. Non-Learning Based Light Field Synthesis

Kubota et al. [19] use focal stacks captured from multiple viewpoints to synthesise intermediate views. They assume that the scene has only two focus regions: a background and a foreground. The inputs for their approach are four images, two images captured by each camera for the background and foreground regions. The drawback of the approach is that it requires images to be captured from two viewpoints, which is a complex setup, and the resultant synthesized image only has only two focal planes.

In their work, Mousnier et al. [20] propose partial light field reconstruction from a focal stack. They use the focal stack images captured by a Nikon camera to estimate the disparity map and an all-in-focus image, and then use the camera parameters to estimate the depth map. They use the depth map and all-in-focus image to synthesise only one set of nine horizontal and nine vertical perspective views, but since the algorithm requires data from the camera parameters, it is difficult to implement the algorithm to check the accuracy against light field sub-aperture images.

Levin et al. [14] also use focal stack images, but, instead of using depth estimation for the synthesis, show that using a focal stack, the 4D light field can be rendered in a linear and depth-invariant manner. They argue that a focal stack is a slice of the 4D light field spectrum; thus, the focal stack directly provides the set of slices that comprise the 3D focal manifold that can be used to construct the 4D light field spectrum. However, their dimensionality gap model is unreliable at depth boundaries, which results in the background pixels leaking into the foreground pixels.

Pérez et al. [25] propose a light field recovering algorithm from its focal stack that is based on the inversion of the Discrete Focal Stack transform. They show that the inversion using the Discrete Focal Stack transform needs many images in the focal stack. They then show practical inversion procedures for general light fields with various types of regularizers, such as L2 regularization of 0th order and 1st order, and L1 isotropic TV regularization. The two main drawbacks of the algorithm proposed by Pérez et al. [25] are that inversion using the Discrete Focal Stack transform requires a large number of images in the focal stack, and they need to use regularization approaches to stabilize the transform.

Zhang et al. [17] in their work use one micro-baseline image pair to synthesize the 4D light field image, where the disparity between the stereo images is less than 5 pixels. They propose that the small-baseline image pair can be captured using vibration in a static camera or by a slight movement of a hand-held camera. There are two limitations of the approach: first, the depth estimation algorithm used reduces in accuracy as the distance between the input views is increased; second, since no information is available to fill in the gaps generated by the difference in the movement of the background and foreground regions in the sub-aperture images, the accuracy of the edge sub-aperture images is reduced considerably compared to the sub-aperture images closer to the central view.

Shi et al. [13] use a sparse set of light field views to predict the views inside the boundary light field images used, but since the approach requires a specific set of sub-aperture views as input data from the light field images, applying the algorithm to different types of data is non-trivial. The approach can be used for applications such as light field compression, but not for light field synthesis as they require a set of sub-aperture views as input data.

### 2.2. Learning-Based Light Field Synthesis

Kalantari et al. [21] propose a two-network learning based light field synthesis approach that uses a sparse set of four corner sub-aperture images. The first network estimates the depth map and then the second network estimates the missing RGB sub-aperture images. Gul et al. [26] propose a three-stage learning-based light field synthesis approach that also uses a sparse set of four corner sub-aperture images. The first stage is the stereo feature extraction network, the second stage is a depth estimation network, and the third stage uses the depth map to warp the input corner view to have them registered with the target view to be synthesized. One drawback of both the proposed algorithm is that capturing the four corner sub-aperture images is not straightforward, and would either require moving the camera manually or a special apparatus with multiple cameras. However, the approach can be used for light field compression as the approach uses corner sub-aperture views as input data.

Srinivasan et al. [22] propose a CNN that estimates the geometry of the scene for a single image and renders the Lambertian light field using that geometry, with a second CNN stage that predicts the occluded rays and non-Lambertian effects. The network is trained on a dataset containing 3300 scenes of flowers and plants captured by a plenoptic camera. However, since the algorithm predicts the 4D light field image using a single image, filling the regions in the sub-aperture image at large discontinuities will only be an approximation as that information is not available from a single image. They extend their network by training it on 4281 light fields of various types of toys including cars, figurines, stuffed animals, and puzzles, but their results show that the images are perceptually not quite as impressive as the images synthesized for the flower dataset [22].

Wang et al. [27] propose a two-stage position-guiding network that uses the left-right stereo pair to synthesize the novel view. They first estimate the depth map for the middle/central view and then check the consistency for the synthesized left and right view. The second CNN is the view rectifying network. They train their network on the Flyingthings3D dataset [28] that contains 22,390 pairs of left-right views and their disparity maps for training and 4370 pairs for testing. The main limitation of the approach is that, since their research focuses on dense view synthesis for light field display, they only generate the central horizontal views and not the entire light field image.

Wu et al. [29] present a “blur restoration-deblur” framework for light field reconstruction using EPIs. They first extract the low-frequency components of the light field in the spatial dimensions using a blur kernel on each EPI slice. They then implement a CNN to restore the angular details of the EPI, and they use a non-blind “deblur” operation on the blurred EPI to recover the high spatial frequencies. In their work, they also show the effectiveness of their approach on challenging cases like microscope light field datasets [29]. The main drawback of their approach is that they need at least three views for both angular dimensions for the initial interpolation, and their framework cannot handle extrapolation.

Yeung et al. [30] propose a learning-based algorithm to reconstruct a densely-sampled light field in one forward pass from a sparse set of sub-aperture views. Their approach first synthesises intermediate sub-aperture images with spatial-angular alternating convolutions using the characteristics of the sparse set of input views, and they then use guided residual learning and stride 2 4D convolutions to refine the intermediate sub-aperture views. They suggest that the proposed algorithm can not-only be used for light field compression but also applications such as spatial and angular super-resolution and depth estimation.

Zhou et al. [31] train a deep network that predicts the Multi-Plane Image (MPI) using an input stereo image pair. A multi-plane image is a set of images where each plane encodes the RGB image and an alpha/transparency map at each depth estimated by the stereo image pair. The MPIs can be considered as a focal stack representation of the scene, predicted using only the stereo image as input. If the stereo baseline is large enough, the parts of the image that are visible due to the lateral shift give information that can be used to fill in the gaps generated by the difference in region depths in the perspective views (horizontal direction). Chao et al. [18] in their work propose a lightweight CNN that uses a single stereo image pair that enforces the left-right consistency constraint on the light fields synthesized from left and right stereo views. The light field synthesized by right and left stereo views is then merged by using a distance-weighted alpha blending operation. However, since the input stereo pair used is only in the horizontal direction, gaps in the vertical perspective views can only be filled by using a prediction model as no information is available in the vertical direction.

## 3. Methodology

The methodology presented in this paper exploits the characteristics of focal stack images and the all-in-focus image to generate a light field image with an angular resolution of 9 × 9. The 9 × 9 resolution is chosen to have the same angular resolution and the images in the dataset; thus, the accuracy of the algorithm can be calculated for the entire light field image. The flow of the algorithm is represented in Figure 1. As shown in Figure 1, the methodology can be divided into three main stages: depth estimation, sub-aperture view synthesis, and RGB and depth map filling for occluded regions.

### 3.1. Depth Estimation

We exploit the characteristics of focal stack images to generate a disparity map that is used to synthesize the light field image. The flow of the depth estimation algorithm is shown in Figure 2. Our algorithm uses the concept of depth from defocus by a one-to-one comparison between each focal stack image and the central all-in-focus image. This estimation approach is also noise-resilient and outperforms the current state-of-the-art benchmark algorithms in the presence of noise [5]. Note that the below only provides an overview of our depth estimation approach; more details are presented in our previous publication [5]. There are other algorithms such as [32,33,34,35] that use stereo matching for depth estimation, but there are two main reasons why we do not compare our depth estimation approach to these algorithms. First, this paper focuses on light field synthesis using focal stacks as they can be used to extract the information used to fill the gaps in the synthesized sub-aperture views. Second, these techniques use sub-aperture views, which are difficult to capture using a 2D camera without physically moving the camera; in contrast, focal stacks can be captured by only adjusting the focal point of the camera lens.

#### 3.1.1. Focal Stack Generation and Image Pre-Processing

Our depth estimation algorithm works with a focal stack image captured by a camera, or generated using the light field image. For the purpose of our work, we use the focal stack images generated using the light field image as we are able to validate the accuracy of the synthesized light field view by using a similarity index metric with the original light field image views.

The focal stack is generated from the light field image by using a shift-sum filter, which is a depth-selective filter that functions like a planar focus. The filter works by shifting the sub-aperture images of the light field image to a common depth and then adding the images together to obtain the 2D image. The average of the shifted sub-aperture pixels values is used for refocusing, as it replicates the blur around depth discontinuities in focal stacks captured by a camera.

To minimize the number of misdetections, the gradient of the image is added to itself to ensure that all the edges and textured regions in the image are well defined in both the central all-in-focus image and the focal stack images. The advantage of gradient addition relies on the fact that, in focal stack images, the textured regions in the image that are in focus maximally contribute to the gradient image, while the out-of-focus objects contribute the least. This pre-processing step ensures that the object boundaries and textured regions are exaggerated in the focal stack images. This drastically reduces the number of misdetections, in turn reducing the dependence on the post-processing steps.

#### 3.1.2. Patch Generation and Comparison

The focal stack images from the previous stage are divided into smaller image patches, with the individual patches then compared with the corresponding patches in the all-in-focus image. For depth estimation, we use the two patches of size 4 × 4 pixels as shown in Figure 3, the squares outlined by red and green lines. The results for depth map accuracy with different window sizes showed that smaller window sizes covered the image regions and boundaries more accurately. We can also reduce the patch size to lower than 4 × 4 pixels; however, experimental tests revealed that using a patch smaller than 4 × 4 pixels does not improve the depth map accuracy and increases the computational time. Figure 3 shows the two 4 × 4 image patches in the red and green squares that are considered for matching. Since we use overlapping windows, we only use the pixels highlighted in the red square and green square for the depth map estimation as shown in Figure 3.

We compared the FFT of the image patches, so we no longer look at individual pixel values when comparing the image patches but a frequency domain representation of those patches, which makes the comparison more noise resilient. Figure 4 and Figure 5 illustrate the proposed approach. Figure 4a shows the central sub-aperture image of the LF image, and Figure 4b,c show a 4 × 4 pixel patch taken from the image and the FFT of the patch, respectively. The depth levels for the depth map depend on the number of images in the focal stack images. In Figure 5, only 9 refocused images are considered from −4 to +4 slope at an interval of 1. It is clearly seen that the FFT of the fifth patch in Figure 5 is the most similar to the FFT of the reference patch.

#### 3.1.3. Depth Estimation and Refining

The estimated depth map still has a few errors, and these are refined in two steps using an iterative approach. Firstly, the histogram of the depth map is checked for the number of pixels that are present at each depth. If the number of pixels at a particular depth falls below a threshold value, those pixels are filled with the maximum likelihood value of the pixels in the depth map at that position, i.e., using the pixel value that occurs the most times in the neighbouring pixels. The second step is similar, but instead of considering individual pixels, the patches are considered. This step checks for any isolated patches in the image that have different surrounding depth patches. Once these patches are isolated, and the patch is then filled with the value of pixels with maximum likelihood in the depth map at that patch position (similar to the above).

### 3.2. Sub-Aperture View Synthesis Using FFT-Shift

Epipolar images are formed by stacking the sub-aperture images in the horizontal and vertical directions, and a representative slice through this 4D block is shown in Figure 6b,c, respectively. The red, green and blue parallelograms show that the slope of the line reflects the depth of the pixels. The pixels that do not appear to move in between the sub-aperture views are seen as a straight line; this is shown by the blue parallelograms in Figure 6b that have a zero slope. The pixels shown by the green parallelograms in Figure 6c that are closer to the camera incline to the right, and the pixels that are further away from the camera incline to the left, as shown by the red parallelograms in Figure 6. The pixels in the sub-aperture view depict the depth as they appear to be moving toward or away from the central view. In turn, if the depth is known, the pixels in all of the sub-aperture views can be filled by using pixels from the central view or the all-in-focus image.

The amount by which the pixels appear to move from the central view in the sub-aperture views is the product of the depth value and the distance of the sub-aperture view from the central view. Since the product of the depth value and the distance of the sub-aperture view from the central view can have small decimal values, we use the frequency domain to fill the sub-aperture views.

The relationship between the image shift in the spatial and frequency domains is shown in Equation (Equation 1), where *x_0_* and *y_0_* are the depth value and *u*, *v* are the sub-aperture location. The amount by which the central view pixels shift to mimic the apparent shift of pixels between sub-aperture views is the product of *x_0_*, *u* and *y_0_*, *v*.
(1)$$\;f\left(x+x_{0},y+y_{0}\right)=F\left(u,v\right)e^{-j2\pi (\frac{u*x_{0}+v*y_{0}}{N})}\;$$

Using the frequency domain to mimic the apparent shift of pixels between sub-aperture views ensures the accuracy of synthesized views at the sub-pixel level. The pixels in the sub-aperture views are thus filled from the minimum depth to the maximum depth in the depth map. As we move through different perspective views, the regions in the image closer to the camera cover the background pixels, and filling the views from the minimum depth values ensures that the regions in the image that overlap the other depths in the sub-aperture views are correctly filled.

### 3.3. RGB and Depth Map Filling of Occluded Regions

Once we fill the pixels in the separate sub-aperture views using the all-in-focus image and depth maps, due to the difference in the depth between different regions, some parts of the image that are not visible in the central view are exposed. This also occurs in the perspective depth map views as shown in Figure 7a.

#### 3.3.1. Depth Map Filling of Occluded Regions

In a depth map, if there are two regions at different depths and a gap is thus created, the region will always be filled by the depth value which is farther away, as the apparent movement of the foreground objects is more than the background objects between sub-aperture views as shown in Figure 7b.

#### 3.3.2. Filling Occluded RGB Regions Using the Focal Stack

Filling the occluded regions in the RGB images is more complex than filling the occluded regions in the depth map, as the depth map values have only two possible values to choose from: the foreground or the background depth values. In our approach, the depth values and the focal stack images are used to estimate the pixel values for the gap generated near depth discontinuities. Due to the defocus blur in focal stack images, when focusing on the background, parts of the background are revealed that are not visible in the all-in-focus image. The amount of blur is also dependent on the depth difference between the foreground and background object. As the pixels in the sub-aperture images also move in accordance with their depth values, the focal stack image reveals the exact amount of information required to fill the gaps, as shown in Figure 8. Figure 8c is obtained by blurring the foreground objects by the amount equivalent to the depth difference between the object and subtracting it from the image focused on the background. However, since the induced blur can only approximate the lens blur, the extracted image region still contains the color tone of the foreground region.

#### 3.3.3. RGB Image Refinement

Figure 9 represents a light field image with an angular resolution of 9 × 9 views. The blue square represents the starting point for the proposed light field synthesis algorithm. We use the all-in-focus image and the estimated depth map to synthesize the central horizontal and vertical sub-aperture views, which in this depth map is represented by the green and yellow squares in Figure 9. Each of the generated central horizontal views and its corresponding depth map are then used to synthesize the sub-aperture views above and below the green squares, while each of the generated central vertical views and its corresponding depth map is then used to synthesize the sub-aperture views to the right and left of the yellow squares. All of the orange squares thus are synthesized using the central horizontal and vertical sub-aperture views, and its depth map is represented by the green and yellow squares. Both of the sub-aperture views generated from the horizontal and vertical views are then averaged as the final light field image.

## 4. Results

### 4.1. Dataset

The results of the proposed algorithm were evaluated on a synthetic 4D light field image dataset [36]. The dataset is widely used to validate depth estimation and reconstruction/synthesis algorithms for light field images as it contains ground-truth disparity and depth maps. The depth range for the synthetic data lies within the range of −4 to +4, and the number of focal stack images can be increased or decreased by reducing or increasing the focus interval between consecutive focal stack images.

The proposed algorithm is compared to three benchmark techniques from Kalantari et al. [21], Chao et al. [18], and Zhang et al. [17]. We have chosen these three techniques because they are state-of-the-art for light field synthesis, and each uses a different approach. Zhang et al. [17] use a micro stereo pair, Chao et al. [18] use a stereo pair with a large baseline, while Kalantari et al. [21] use the four corner sub-aperture views for light field synthesis. The Structural Similarity Index Measure (SSIM) [37] and Peak-Signal-to-Noise-Ratio (PSNR) metrics were used for evaluation.

For Kalantari et al. [21], we have used the trained network used by the authors to synthesize the light field images, and for Chao et al. [18], we trained the network using the code provided by the authors. For the SSIM metric, we compare the top-left sub-aperture views for the algorithms that synthesize a 9 × 9 or 8 × 8 light field image. We can use any of the four corner images for evaluation. We use the corner views for evaluation as they show the maximum parallax from the central view. For the algorithm that only synthesizes the horizontal sub-aperture views, we use the central left-most horizontal view for evaluation. For the PSNR metric, we convert the sub-aperture light field image to the lenslet view and calculate the PSNR.

Kalantari et al. [21] synthesize the light field image using four corner sub-aperture images with an angular resolution of 8 × 8, whereas we synthesize the light field image with an angular resolution of 9 × 9. We thus evaluate the results for comparison by using only the inner-most 8 × 8 sub-aperture views. Chao et al. [18] use two horizontal corner sub-aperture images with an angular resolution of 9 × 9. As they train their network on 20 images from the dataset, only four light field images remain for testing, so we evaluate the average PSNR and SSIM for the four test images. The algorithm proposed by Zhang et al. [17] only synthesizes horizontal sub-aperture views using two micro baseline stereo pairs, so we evaluate the results for only the horizontal views.

The images in Figure 10 show the synthesized leftmost horizontal view and the SSIM error map for three dataset images. The pixels in the SSIM map most similar to the ground-truth sub-aperture view appear white, where the similarity index is close to 1. In contrast, the regions in the image least similar appear red, where the similarity index is close to 0, as shown in Figure 11.

For depth estimation, we use a depth from defocus technique that uses the focal stack images and an all-in-focus image to estimate the depth map. As the number of images in the focal stack govern the resolution of the depth map, the accuracy of the synthesized light field images reduces as the number of focal stack images reduces. Table 1 shows the average PSNR and SSIM for the proposed algorithm for different numbers of focal stack images used for light field synthesis for all images in the dataset [36]. It should be noted that, for the sake of generalization, the focal stack images are captured over the entire depth range for the synthetic data; that is, from −4 to +4 irrespective of the depth range of individual images in the dataset. Thus, even though the number of focal stack images for light field synthesis are 41, 21, and 17 in Table 1, the focal stack images that have regions in focus are less than the total number of images in the focal stack. We chose the number of focal stack images as 41, 21 and 17, as these values generate consecutive focal stack images at a depth difference of 0.2, 0.4, and 0.5, respectively, for the depth range of −4 to +4. Furthermore, only the images that have regions in focus are constituted in the estimation of the depth map. For a focal stack with five images marked with a ‘*’ in Table 1, the focal stack images are captured between the maximum and minimum depth values for the depth range of each image.

### 4.2. Quantitative Analysis Synthetic Images

The quantitative results are divided into two parts, as the method of Zhang et al. [17] only generates the central horizontal sub-aperture views. Figure 10 and Table 2 show the visual comparison and quantitative results for the leftmost horizontal view for the Boxes, Cotton and Dino image for Zhang et al. [17], Kalantari et al. [21] and Chao et al. [18] using the PSNR and SSIM metrics. Table 3 shows the average PSNR and SSIM for Zhang et al. [17] and the proposed algorithm for the central horizontal views for all the images in the dataset [36]. Table 4 shows the average PSNR and SSIM for Kalantari et al. [21] and the proposed algorithm for 8 × 8 views for all the images in the dataset [36]. For Chao et al. [18], since 20 images are used for training the network, Table 5 shows the average PSNR and SSIM for the four test images in the dataset [36].

#### 4.2.1. Quantitative Analysis for Central Leftmost Horizontal Sub-Aperture View

The ‘Boxes’ image in Figure 10 consists of a crate with books in the foreground and bags in the background. None of the algorithms can accurately synthesize the fine criss-cross pattern on the crate, as shown by the yellow and red regions in the SSIM error map of Figure 10a. While our algorithm can still maintain the criss-cross pattern of the crate, the results for Zhang et al. [17] show distortion for this foreground pattern. For Kalantari et al. [21], the criss-cross pattern is invisible in some regions in the synthesized view, whereas, for Chao et al. [18], the pattern appears to be doubled. The results for Zhang et al. [17] also show distortion around the edges of the crate and the box placed on the crate. For the view synthesized by Kalantari et al. [21], the thread pattern on the box placed on the crate is synthesized inaccurately, and the pattern appears twice in some regions. For Chao et al. [18], the boundaries of the bags in the background appear to be shifted and superimposed on the image, making it appear blurred.

The ‘Cotton’ image in Figure 10b is relatively simple as it only consists of a statue and a plain colored wall in the background. Our results show that, except at depth discontinuities, our algorithm can synthesize the image accurately for all the other regions in the image: for PSNR and SSIM in Table 2, the accuracy is 44.99 and 0.9953, respectively. For Zhang et al. [17], the synthesized image and SSIM error map show that, near the head of the statue on the left side, the boundaries are pixelated, and on the right side, part of the head is stretched. It is also be seen in the SSIM error map in Figure 10b that most of the depth discontinuities within the statue and the shadow regions in the figure are also incorrectly synthesized. For Kalantari et al. [21], we can see from Figure 10b that the area within the statue is synthesized accurately, except for the regions where a shadow is cast on the statue. Some texture near the top of the head is missing, represented by the red area in the SSIM error map. The image background region is synthesized inaccurately, shown by the yellow areas in the SSIM error map. For Chao et al. [18], even though the SSIM score is 0.9357, the structure of the eyes, nose, and hair appear to be perceptually shifted and superimposed on the image, making it appear blurred near those regions in the figure.

The ‘Dino’ image in Figure 10c is relatively complex as it consists of a textured wooden background, the cast of a dinosaur shadow on the wall, and wooden toys and boxes. Our results show a similar trend as the ‘Boxes’ and ‘Cotton’ images, where our algorithm can synthesize the image accurately for all the regions in the image except at depth discontinuities: the PSNR and SSIM results show the accuracy to be 38.42 and 0.9901, respectively, as shown in Table 2. For Zhang et al. [17], the resultant image and SSIM error map show that most of the depth discontinuities in the figure are also incorrectly synthesized. It can also be seen in the synthesized view in Figure 10c that the open wooden shelves on the left and the wooden boxes on the right have jagged edges. For Kalantari et al. [21], we can see that the synthesized view from Figure 10c has no visual errors. Still, the SSIM error map shows that most regions in the image appear yellow, implying that the accuracy of the synthesized views compared to the actual light field view is between 80–90% (as indicated by the color bar in Figure 11). For Chao et al. [18], we see a similar trend as the ‘Boxes’ and ‘Cotton’ images, where the synthesized view has regions in the image that appear to be shifted and superimposed on the image, making it appear blurred near those regions in the figure. This effect is visible near the dinosaur shadow on the wall, and the wooden toys near the bottom of the image, where the image appears blurred.

#### 4.2.2. Quantitative Analysis for Top-Leftmost Sub-Aperture View

Since Kalantari et al. [21] and Chao et al. [18] synthesise the light field image with angular resolution on 8 × 8 and 9 × 9, respectively, we can compare the appropriate corner sub-aperture view to measure the algorithm’s accuracy. Figure 12 and Table 6 show the visual comparison and quantitative results for the top left sub-aperture view. The results for our algorithm for the top corner sub-aperture views reduce in accuracy compared to the horizontal views. As our algorithm uses depth maps to synthesize the sub-aperture views, the error in the depth map for horizontal views is amplified for the top and bottom sub-aperture views, which reduces the synthesized image accuracy for the top and bottom views. For all images in the dataset, the accuracy is reduced for the top left view compared to the horizontal view for our algorithm: the PSNR reduces from 33.55 to 31.24 and the SSIM reduces from 0.9713 to 0.9525. For Chao et al. [18], the accuracy for the top corner sub-aperture views also reduces compared to their synthesized horizontal views, but the drop in accuracy is quite significant for SSIM. For the four test images evaluated, the accuracy for the top left view compared to the horizontal view in terms of PSNR reduces from 23.74 to 21.8, whereas the SSIM reduces from 0.9093 to 0.7902. In contrast, for Kalantari et al. [21], as the input images used are the four corner sub-aperture views, the accuracy of the synthesized views reduces as we move towards the central views from the four corner views. For all images in the dataset, the average reduction in accuracy for the horizontal view compared to the top-left view is a PSNR reduction from 19.21 to 18.62 and SSIM reduction from 0.8872 to 0.8071.

For the ‘Boxes’ image in Figure 12a, as in the case of horizontal views, the top left sub-aperture view also struggles with the fine criss-cross pattern on the crate, as seen in the yellow and red regions in the SSIM error map. For our algorithm, the depth map inaccuracies cause some regions to be synthesized incorrectly. This effect is visible above the box near the upper left part of the image, where the boundaries of the bags are shifted slightly to the right. The results for Kalantari et al. [21] and Chao et al. [18] show errors in similar regions as seen in Figure 12a, which is near the top edge of the box and the crate.

For the ‘Cotton’ image in Figure 12b, our results show similar high accuracy for the top left view as the horizontal view, where the algorithm can synthesize the image accurately for all the regions except at depth discontinuities. The PSNR and SSIM results shown in Table 6 show the accuracy to be 43.07 and 0.9925, respectively. For Kalantari et al. [21], we can see from Figure 12b that the background region in the image is synthesized inaccurately, as shown by the orange regions in the SSIM error map. Areas in the image with shadows near the neck and shoulder are also inaccurately synthesized. For Chao et al. [18], similar to the horizontal view, the structure of the eyes, nose, and hair appear to be shifted and superimposed on the image, making it appear blurred near those regions in the figure.

In the ‘Dino’ image in Figure 12c, our algorithm accurately synthesizes the image for all the regions in the image except at depth discontinuities, where the SSIM error map appears yellow. For Kalantari et al. [21], we can see that the synthesized view from Figure 12c has no visual errors with an SSIM of 0.9420, but this high accuracy is also because the corner sub-aperture views are used as input images for the synthesis. Still, the SSIM error map shows that most regions in the image appear yellow, implying that the accuracy of the synthesized views compared to the actual light field view is between 80–90% (see color bar in Figure 11). For Chao et al. [18], near the dinosaur shadow on the wall and the wooden toys near the bottom of the image, parts of the image appear to be shifted and superimposed on the image, making it appear blurred near those regions in the figure.

In the ‘Sideboard’ image in Figure 12d, our synthesized view shows errors in two regions in the image. As our depth estimation algorithm cannot distinguish the depth for thin objects, the ceiling wires on which the lights hang from are incorrectly synthesized. The other error is near the bottom of the image, where the legs of the sideboard appear distorted. Again, this is because the depth estimation algorithm misdetected the depth of the legs. For Kalantari et al. [21], we can see that the sideboard and the objects placed on the sideboard are accurately synthesized as these regions in the SSIM error map appear white, whereas all other regions appear yellow. For Chao et al. [18], the pattern of the wall and the objects placed on the sideboard appear to be blurred, which is again because it appears as a shifted image superimposed on the image.

### 4.3. Quantitative Analysis for Real Light Field Image

To test the accuracy of our algorithm on real light field images, we use the 30-scene dataset [21]. We compare the accuracy of our algorithm with Wu et al. [29], Yeung et al. [30] and Kalantari et al. [21]. Table 7 shows the PSNR and SSIM results averaged over all 30 images in the dataset. We synthesize 7 × 7 sub-aperture views for the real light field images using the focal stack images and the central all-in-focus image. The results for Wu et al. [29], Yeung et al. [30] and Kalantari et al. [21] have been taken from the results presented by these authors in their work. For the real images, the depth values range from +2 to −2, as opposed to synthetic images, which have depth values ranging from +4 to −4. Since our algorithm is a non-learning-based approach, the only change we make to synthesize real light field images is to change the depth range, which shows the flexibility of our approach. It can be seen from the results shown in Table 7 that our approach produces on par results in terms of the similarity index(SSIM) but reduces in accuracy in terms of the PSNR values. The reduction is because Wu et al. [29] takes 3 × 3 input images and only interpolates one image between their input views. Kalantari et al. [21] use the corner images as input and interpolate all the internal views, while Yeung et al. [30] in their 2 × 2–8 × 8 set-up only extrapolate two views in both directions, while the other views are interpolated within the baseline of the input images. On the other hand, we only use the central image as input and extrapolate three images in both directions to synthesize a 7 × 7 light field image. We mainly use the focal stack image and the all-in-focus image as input instead of sub-aperture views because sub-aperture views are comparatively more difficult to capture. In addition, while extrapolating to synthesize the views using sub-aperture images, we do not have any information to fill the occluded regions.

Figure 13 shows the visual comparison for the left-topmost sub-aperture image with the ground truth for four images from the 30-Scenes dataset [21]. Table 8 shows the PSNR and SSIM results for the four images shown in Figure 13 from the 30-Scenes dataset [21]. We have chosen these images as there is a significant depth difference between the foreground and background regions in these images. In Figure 13a, the tree’s bark covers part of the road and the car. It can be seen from the magnified images that the bark in the synthesized image shows no blurring around the edges near the road or the car, as seen in the red and green magnified images, respectively. The flower scene in Figure 13b consists of plants and trees in the foreground and cars, houses and a man in the background. The magnified images in Figure 13b show that the edges of the leaves are sharp, and even the bark with the house in the background is synthesized correctly. However, closer inspection of the image reveals that just to the left of the green magnification window, the edges of the window on the house in the background slant a little to the right. An error of the depth map causes this abnormality in the synthesized view. A similar aberration can be seen in Figure 13c in the magnified green window, but this irregularity is due to the incorrect filling of the occluded region of the image. This irregularity can also be seen in the SSIM map, highlighted by the dark red spots. In Figure 13d, the red magnification window shows no blurring effect near the seahorse’s snout, but if we look closely at the green magnification window, we notice that the gap between the seahorse and the chair handle is less than seen in the ground truth image. This is again due to an error with the depth map, as the car in the background or the seahorse is estimated at a slightly incorrect depth, causing the objects to appear closer.

### 4.4. Results Overview

It is clear from the visual comparison shown in Figure 10, Figure 12 and comparative quantitative results presented in Table 2 to Table 6 that our proposed algorithm outperforms the three algorithms for both PSNR and SSIM metrics. One main disadvantage of Kalantari et al. [21] is that they use four corner sub-aperture views for synthesis, and it is not easy to capture the corner views without moving the camera. Chao et al. [18] uses a large baseline horizontal stereo pair and interpolates the other horizontal views within that baseline. Still, as no information is available for the extrapolated vertical views, the algorithm’s accuracy reduces for the corner sub-aperture views. Furthermore, for our algorithm, the final resolution of the light field image mainly depends on the resolution of the central all-in-focus image. The precision of the depth map only ensures parallax accuracy in the sub-aperture views. Thus, even if the depth map precision is reduced, this will only reduce the amount of parallax of the synthesized light field image. Still, the resolution of the light field will correspond to the central view’s resolution.

Wu et al. [29] and Kalantari et al. [21] only interpolate images within the baseline of the input image to synthesize the internal views. Yeung et al. [30] in their 2 × 2–8 × 8 set-up only extrapolate two views in each direction, while the other views are interpolated within the baseline of input images. Since these algorithms interpolate most of the synthesized views between the baseline of the input views, they achieve higher accuracy than our approach, but as these algorithms require sub-aperture views as input, these algorithms are not practical for light field synthesis using 2D cameras. In contrast, focal stack images can be captured relatively easily as we do not need to use any additional equipment to move the camera to capture different viewpoints; instead, we only need to change the camera’s focal point.

## 5. Future Work

In our approach proposed here, the number of depth levels in the depth map is dependent on the number of images in the focal stack. Thus, with fewer images in the focal stack, if an object in the image has two or more depth levels, the object shows an abrupt discontinuity in the synthesized view, as shown in Figure 14 (highlighted by the red squares). In our future work, we intend to use the focal stack images to estimate the amount of blur for the same defocus regions between consecutive focal stack images and use that information to increase the depth levels of the depth map. Increasing the depth levels using fewer focal stack images will reduce the effect of abrupt discontinuities for objects with two or more depths in the synthesized view, increasing the synthesized view accuracy with fewer focal stack images. In our future work, for light field synthesis using focal stack images captured by a 2D camera, we intend to compare our algorithm accuracy with RVS and VSRS view synthesis algorithms [38] that use DIBR.

Some commercial cameras such as Lumix [39] and Olympus [40] already have a feature called focus stacking that can take high-resolution focal stack images and merge them to create a sharper all-in-focus image. These cameras also allow the user to save the individual focal stack images in the raw format. Thus, with further development, our algorithm can help to enable light field creation. In our future work, we intend to use these focal stack images and all-in-focus images to synthesize the light field image with an angular resolution of 15 × 15 with a high spatial resolution of the individual sub-aperture views. We also intend to capture the focal stack with a 2D camera and align the light field camera with the 2D camera to capture a light field of the same scene and use that as reference views to check the accuracy of our approach.

## 6. Conclusions

We propose a light field synthesis algorithm that uses the focal stack images and the all-in-focus image to synthesize high-accuracy light field images with varying sizes of focal stacks as input with an angular resolution of 9 × 9. We fill the occluded regions with the information recovered from the focal stack images. The depth map and the all-in-focus image synthesize the sub-aperture views and their corresponding depth maps by mimicking the apparent shifting of the central image according to the depth values. We ensure sub-pixel accuracy for small depth values by using the frequency domain to mimic the apparent movement of the regions at different depths in the sub-aperture view. Our algorithm’s accuracy is compared with three state-of-the-art algorithms: one non-learning and two CNN-based approaches. The results show that our algorithm outperforms all three in terms of PSNR and SSIM evaluation metrics. We also show that, if the depth levels in the image are known, we can synthesize high-accuracy light field images with just five focal stack images.

## Figures and Tables

**Figure 1 sensors-23-02119-f001:**
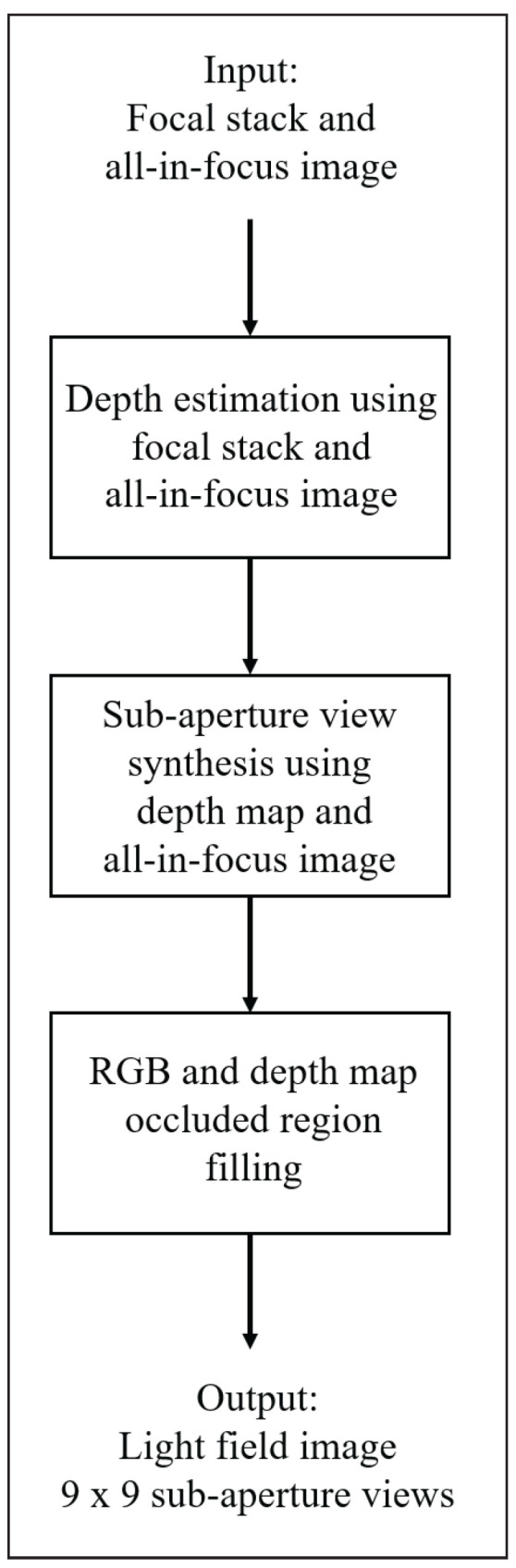
Light field synthesis algorithm flow.

**Figure 2 sensors-23-02119-f002:**
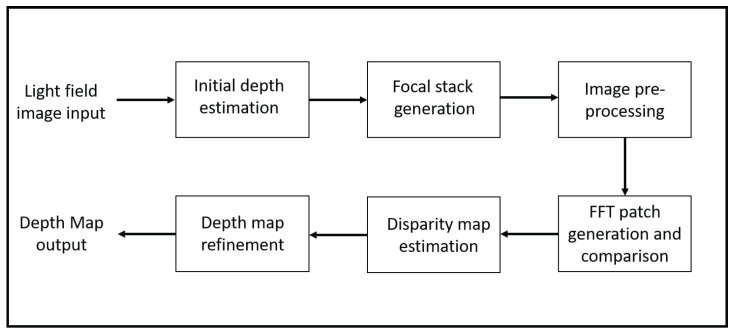
Depth estimation algorithm flow.

**Figure 3 sensors-23-02119-f003:**
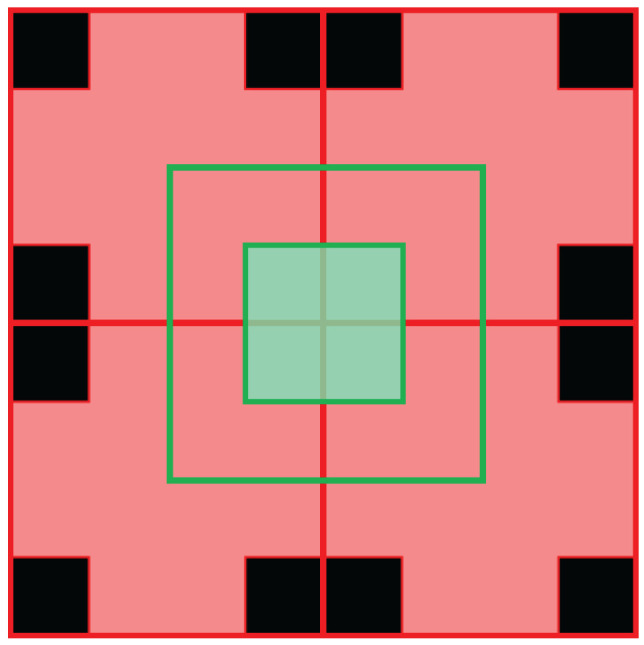
The red and green squares are the two overlapping 4 × 4 pixel patches used to cover the entire image. As the patches overlap, only the highlighted red and the green pixels from the red and green squares are used to estimate the depth.

**Figure 4 sensors-23-02119-f004:**
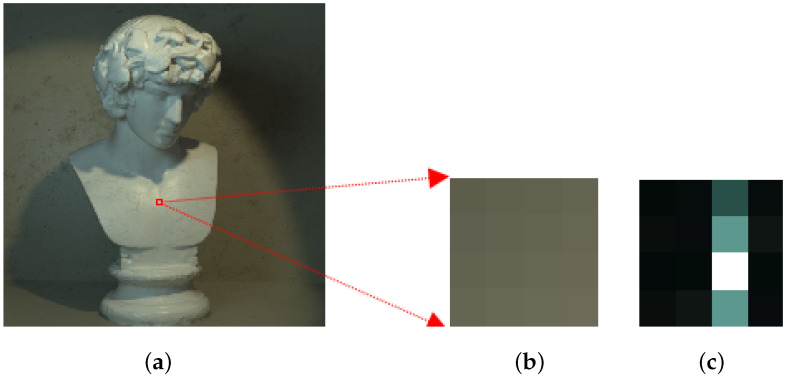
(**a**) Central sub-aperture image of a LF image, (**b**) a magnified 4 × 4 RGB image patch and (**c**) FFT of the image patch.

**Figure 5 sensors-23-02119-f005:**
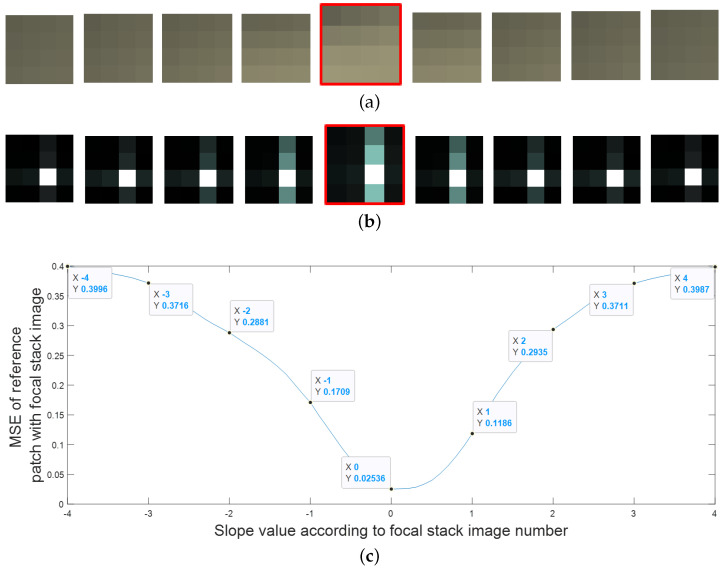
(**a**) The RGB image patch and (**b**) the FFT patch at different focal lengths. The patch with the red boundary is the closest match to the reference patch in Figure 4. (**c**) The graph shows the MSE values for the central image in Figure 4, with the corresponding focal stack image patch.

**Figure 6 sensors-23-02119-f006:**
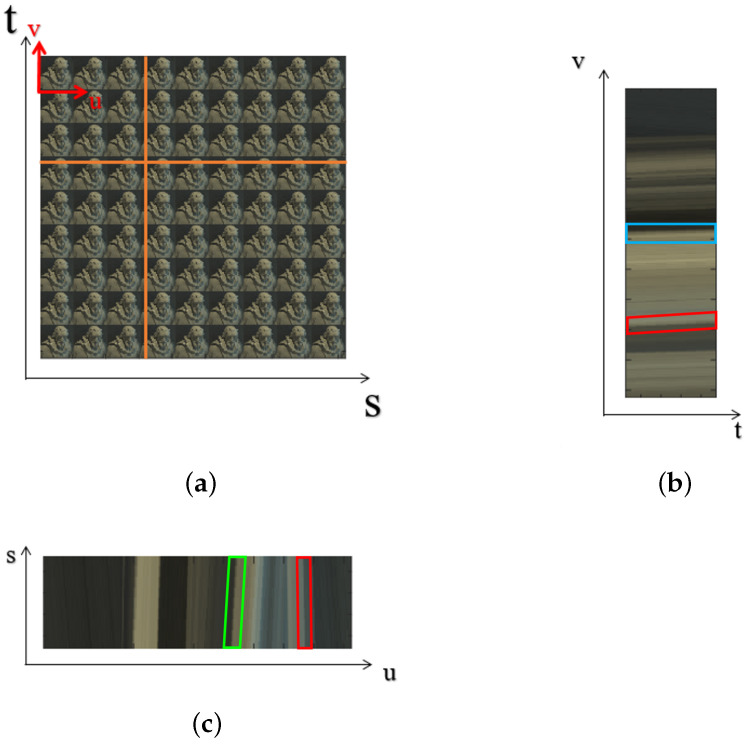
(**a**) The sub-aperture image view; (**b**) the EPI for the vertical line represented in (**a**); (**c**) the EPI for the horizontal line represented in (**a**).

**Figure 7 sensors-23-02119-f007:**
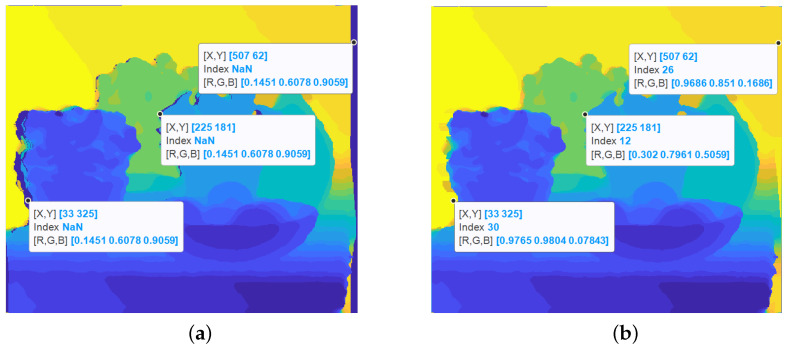
(**a**) Gaps generated in the depth image due to occlusion, and (**b**) Depth map after the gaps are filled.

**Figure 8 sensors-23-02119-f008:**
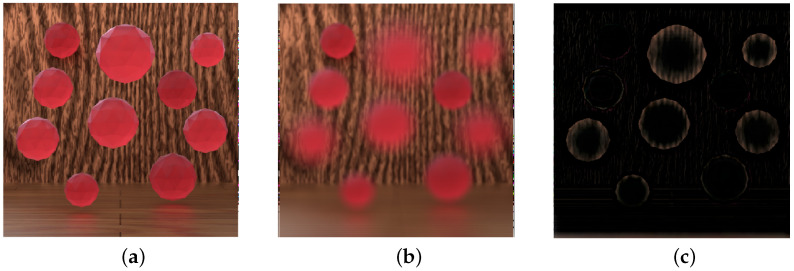
(**a**) The all-in-focus image, (**b**) the image in the focal stack focused on the background, and (**c**) background region minus the effect of the foreground object blur.

**Figure 9 sensors-23-02119-f009:**
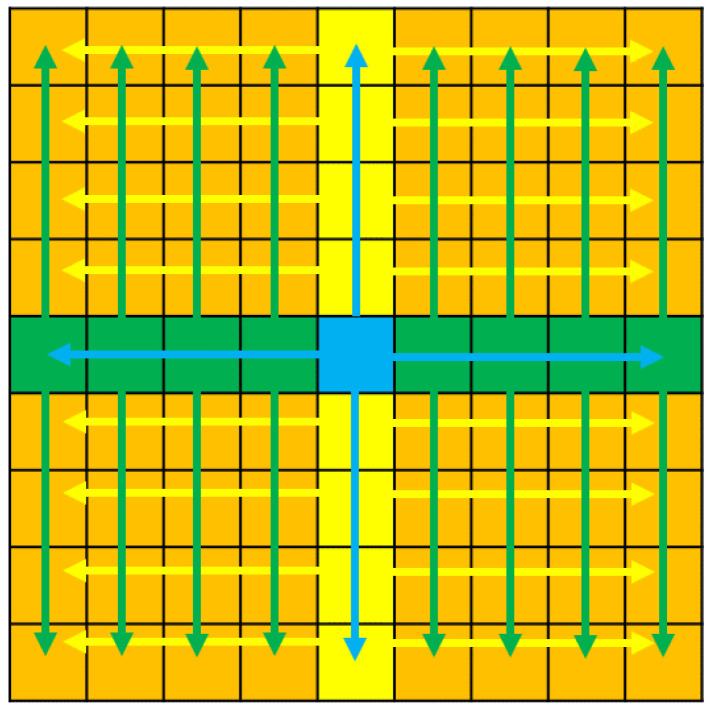
The figure shows the order in which the sub-aperture images are generated.

**Figure 10 sensors-23-02119-f010:**
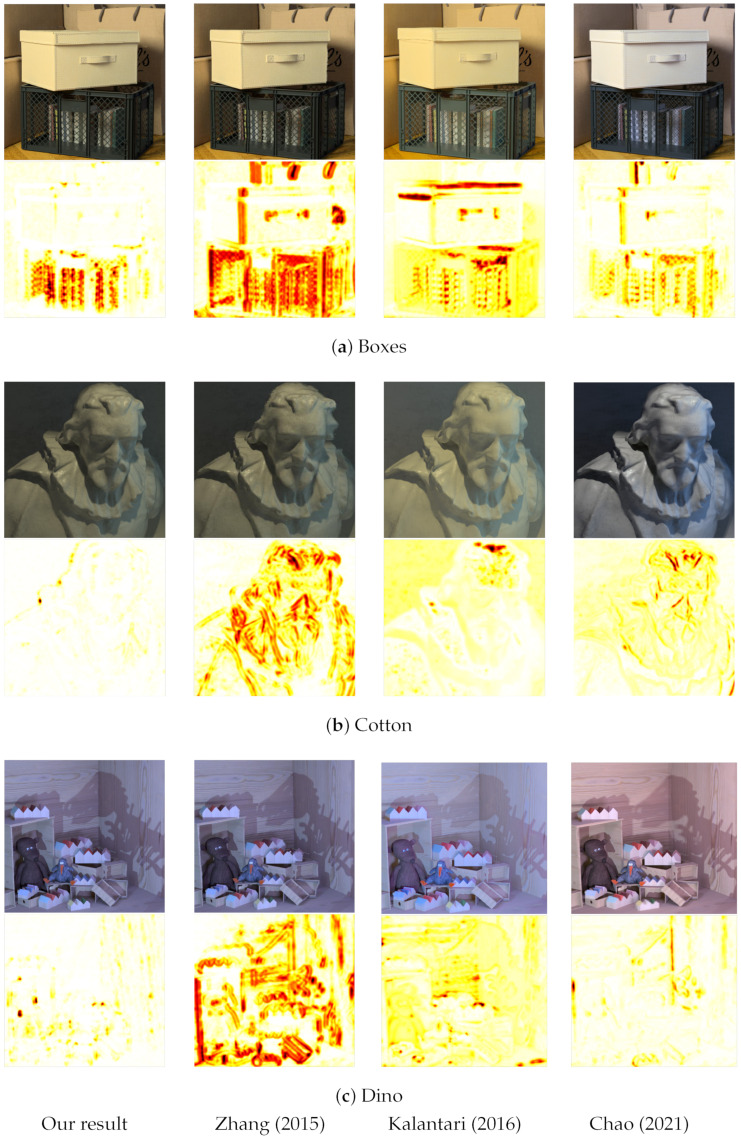
Visual comparison for the ‘Boxes’, ‘Cotton’, and ‘Dino’ image synthesized leftmost horizontal sub-aperture view and the SSIM with the ground-truth sub-aperture view for the proposed algorithm, Zhang et al. [17], Kalantari et al. [21] and Chao et al. [18].

**Figure 11 sensors-23-02119-f011:**
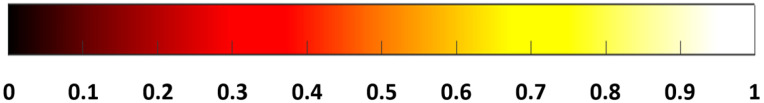
SSIM error map colorbar.

**Figure 12 sensors-23-02119-f012:**
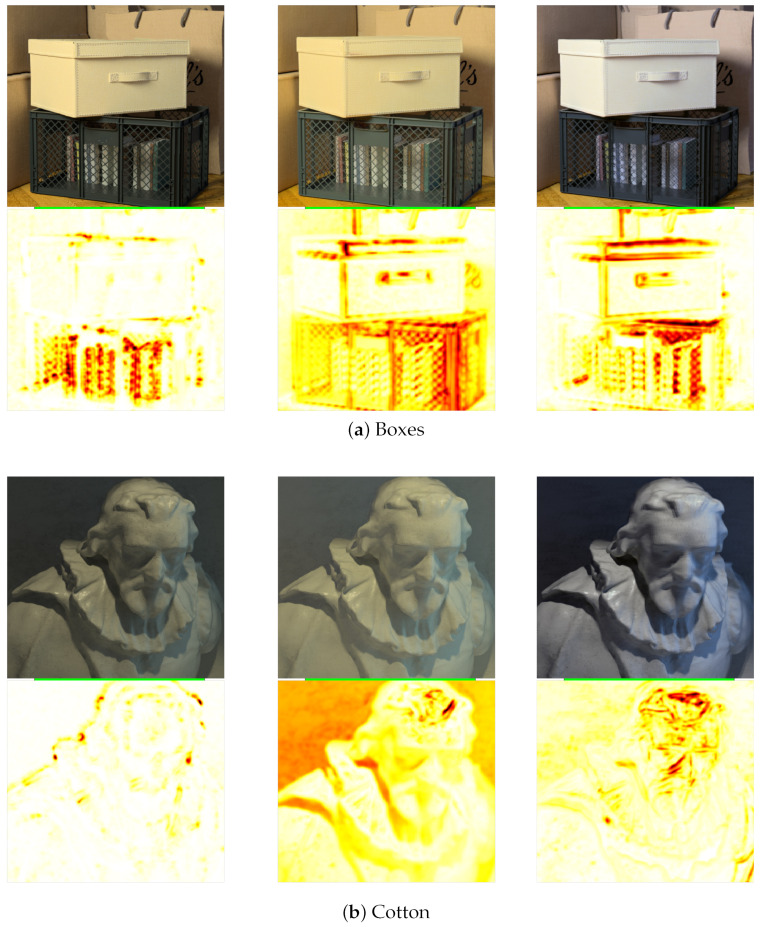
Visual comparison for the ‘Boxes’, ‘Cotton’, ‘Dino’ and ‘Sideboard’ image synthesized top left sub-aperture view and the SSIM with the ground-truth sub-aperture view for the proposed algorithm, Kalantari et al. [21] and Chao et al. [18].

**Figure 13 sensors-23-02119-f013:**
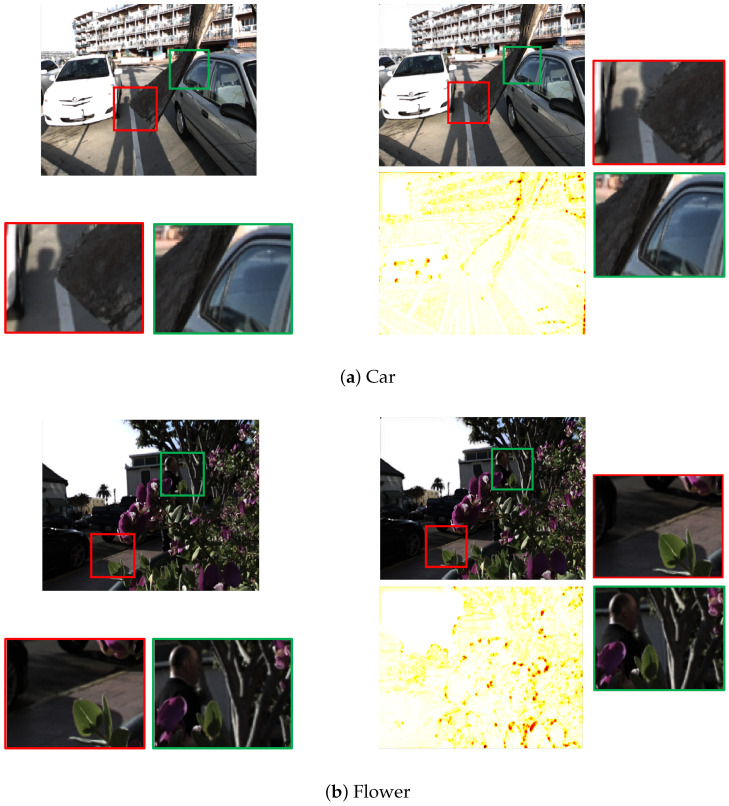
Visual analysis for the ‘Car’, ‘Flower’, ‘Leaves’ and ‘Seahorse’ images’ synthesized leftmost horizontal sub-aperture view and the SSIM map with the ground-truth sub-aperture view for the proposed algorithm.

**Figure 14 sensors-23-02119-f014:**
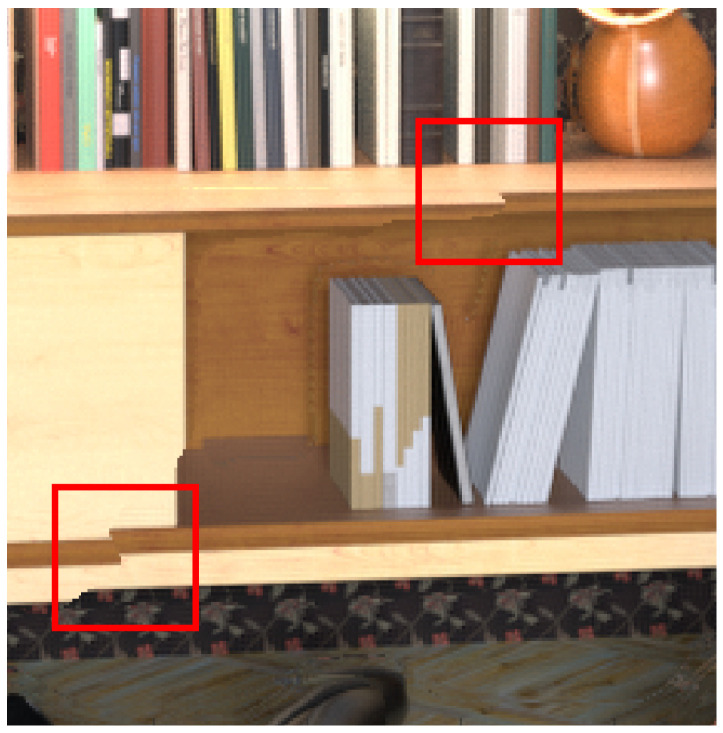
Error in light field synthesis using fewer images in the focal stack.

**Table 1 sensors-23-02119-t001:** Average PSNR and SSIM for all images in the dataset using focal stacks of varying sizes. For focal stacks with five images, the focal stack images used are captured between the maximum and minimum depth value for the depth range of each image.

	41	21	17	5 *
PSNR	33.69	32.23	31.5	29.63
SSIM	0.9588	0.9461	0.9395	0.9052

**Table 2 sensors-23-02119-t002:** Quantitative comparison for leftmost central horizontal synthesized view comparison with different algorithms for images shown in Figure 10.

	Boxes	Cotton	Dino
		Our result	
PSNR	28.41	44.99	38.42
SSIM	0.9313	0.9953	0.9901
		Zhang et al. [17]	
PSNR	21.50	25.72	23.15
SSIM	0.6362	0.7390	0.6801
		Kalantari et al. [21]	
PSNR	19.74	17.21	19.16
SSIM	0.8139	0.8902	0.9207
		Chao et al. [18]	
PSNR	27.34	24.85	19.67
SSIM	0.9260	0.9357	0.9355

**Table 3 sensors-23-02119-t003:** For comparison with Zhang et al. [17], we evaluate the average PSNR and SSIM for only horizontal for all images in the dataset as the algorithm only synthesizes horizontal views.

	Our Result	Zhang et al. [17]
PSNR	32.23	21.1
SSIM	0.9313	0.7592

**Table 4 sensors-23-02119-t004:** For comparison with Kalantari et al. [21], we evaluate the average PSNR and SSIM for all images in the dataset.

	Our Result	Kalantari et al. [21]
PSNR	33.69	18.6
SSIM	0.9588	0.8834

**Table 5 sensors-23-02119-t005:** For comparison with Chao et al. [18], we evaluate the average PSNR and SSIM for four test images in the dataset as the remaining 20 images are used to train their network.

	Our Result	Chao et al. [18]
PSNR	38.08	20.02
SSIM	0.9672	0.8901

**Table 6 sensors-23-02119-t006:** Quantitative comparison for top left synthesized view comparison with different algorithms for images shown in Figure 12.

	Boxes	Cotton	Dino	Sideboard
Our result				
PSNR	27.52	43.07	35.92	25.83
SSIM	0.9137	0.9925	0.9808	0.9431
Kalantari et al. [21]				
PSNR	19.77	17.40	19.44	20.04
SSIM	0.8417	0.9150	0.9420	0.9324
Chao et al. [18]				
PSNR	23.72	24.63	19.32	19.51
SSIM	0.7938	0.8955	0.8614	0.6102

**Table 7 sensors-23-02119-t007:** Comparison with Wu et al. [29], Yeung et al. [30] and Kalantari et al. [21] for the 30-scenes dataset.

30 Scenes Dataset	Our Result	Wu et al. [29]	Yeung et al. [30]	Kalantari et al. [21]
PSNR	36.24	41.02	40.93	37.50
SSIM	0.9922	0.9968	0.98.27	0.97

**Table 8 sensors-23-02119-t008:** PSNR and SSIM for the four images shown in Figure 13 from the 30-scenes dataset.

30 Scenes Dataset	Car	Flower	Leaves	Seahorse
PSNR	30.02	30.98	28.27	31.04
SSIM	0.9921	0.9877	0.9786	0.9923

## Data Availability

The results presented in the paper are available at https://github.com/rishabhsharma27/Light_field_synthesis_results (accessed on 23 September 2022).

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
