# Peer review of "Light Field View Synthesis Using the Focal Stack and All-in-Focus Image"

_sensors, 2023, doi:10.3390/s23042119_

Round 1

Reviewer 1 Report

This paper introduces a light field view synthesis method using images focused on different depth planes (focal stack) as well as an all-in-focus image. The method first uses focal stack to estimate a depth map, then synthesizes novel views using the estimated depth and all-in-focus center view. 

- In the experiments, the results produced by Kalantari (2016) and Chao (2021) appear severe color aberration. The authors shall check that.

- Some detailed descriptions are missing. For examples, in Sec. 3.1.3 the patch comparison is not introduced, in Sec. 3.2 the FFT-shift method for sub-aperture view synthesis is not introduced.

- There are only two articles [5, 18] publised in the last 3 years. More recent researches about light field view synthesis and reconstruction should be reviewed.

- In Line. 67 Sec. 1, the authors states that the input data (sub-aperture images) is not easy to capture. Is it because all-in-focus images are difficult to obtain? The proposed method also uses all-in-focus view for depth estimation and view synthesis. In addition, in Line. 69, I think light field (view) synthesis can also be considered as increasing the angular resolution of a light field.

- In the Abstract, the authors indicate that "light field reconstruction and synthesis algorithms are essential for improving the lower spatial resolution for hand-held plenoptic cameras." Why? While I think the major task for light field reconstruction and view synthesis is to increasing the angular resolution.

Reviewer 2 Report

This paper proposes a light field synthesis algorithm which applies the focal stack images and the all-in-focus image to synthesize high-accuracy light field images with varying sizes of focal stacks as input. Please address the following questions.

1.       How was the accuracy of the synthesized light field view validated in this study?

2.       Would the method proposed by this manuscript be able to generate high resolution image? If so, is there a limitation on the resolution that can be generated?

3.       What are the drawbacks of the image synthesis approach presented in this paper?

4.       How likely is the approach be able to be applied by commercially available cameras in the future?
